# Single-Cell Transcriptomic Analysis of Tumor-Derived Fibroblasts and Normal Tissue-Resident Fibroblasts Reveals Fibroblast Heterogeneity in Breast Cancer

**DOI:** 10.3390/cancers12051307

**Published:** 2020-05-21

**Authors:** Aimy Sebastian, Nicholas R. Hum, Kelly A. Martin, Sean F. Gilmore, Ivana Peran, Stephen W. Byers, Elizabeth K. Wheeler, Matthew A. Coleman, Gabriela G. Loots

**Affiliations:** 1Physical and Life Sciences Directorate, Lawrence Livermore National Laboratory, Livermore, CA 94550, USA; sebastian4@llnl.gov (A.S.); hum3@llnl.gov (N.R.H.); martin249@llnl.gov (K.A.M.); gilmore24@llnl.gov (S.F.G.); coleman16@llnl.gov (M.A.C.); 2School of Natural Sciences, University of California Merced, Merced, CA 95343, USA; 3Georgetown-Lombardi Comprehensive Cancer Center, Department of Oncology, Georgetown University Medical Center, Washington, DC 20007, USA; ip62@georgetown.edu (I.P.); byerss@georgetown.edu (S.W.B.); 4Engineering Directorate, Lawrence Livermore National Laboratory, Livermore, CA 94550, USA; wheeler16@llnl.gov; 5Department of Biochemistry and Molecular Medicine, University of California Davis, Sacramento, CA 95817, USA

**Keywords:** cancer-associated fibroblasts, breast cancer, mammary fat pad, gene expression profiling, scRNA-seq, CAF heterogeneity, normal fibroblasts, myofibroblasts, inflammatory fibroblasts, pancreatic cancer

## Abstract

Cancer-associated fibroblasts (CAFs) are a prominent stromal cell type in solid tumors and molecules secreted by CAFs play an important role in tumor progression and metastasis. CAFs coexist as heterogeneous populations with potentially different biological functions. Although CAFs are a major component of the breast cancer stroma, molecular and phenotypic heterogeneity of CAFs in breast cancer is poorly understood. In this study, we investigated CAF heterogeneity in triple-negative breast cancer (TNBC) using a syngeneic mouse model, BALB/c-derived 4T1 mammary tumors. Using single-cell RNA sequencing (scRNA-seq), we identified six CAF subpopulations in 4T1 tumors including: 1) myofibroblastic CAFs, enriched for α-smooth muscle actin and several other contractile proteins; 2) ‘inflammatory’ CAFs with elevated expression of inflammatory cytokines; and 3) a CAF subpopulation expressing major histocompatibility complex (MHC) class II proteins that are generally expressed in antigen-presenting cells. Comparison of 4T1-derived CAFs to CAFs from pancreatic cancer revealed that these three CAF subpopulations exist in both tumor types. Interestingly, cells with inflammatory and MHC class II-expressing CAF profiles were also detected in normal breast/pancreas tissue, suggesting that these phenotypes are not tumor microenvironment-induced. This work enhances our understanding of CAF heterogeneity, and specifically targeting these CAF subpopulations could be an effective therapeutic approach for treating highly aggressive TNBCs.

## 1. Introduction

Tumors are heterogeneous cellular entities in which progression depends on the dynamic crosstalk between cancer cells and other cells present in the stromal microenvironment [1]. The stroma is composed of supporting cells including fibroblasts, vascular endothelial cells, pericytes, mesenchymal stem cells and various types of immune cells. These cells are surrounded by fibrous structural proteins that comprise a dense extracellular matrix [2]. Cancer-associated fibroblasts (CAFs) are a predominant stromal cellular component in most solid tumors including breast, prostate and pancreatic cancers [3,4]. Previously it has been shown that resident tissue fibroblasts, bone marrow-derived mesenchymal stem cells, hematopoietic stem cells, adipocytes and endothelial cells can all give rise to CAFs, each by different mechanisms [5,6]. In addition, CAFs may arise directly from the cancer cells themselves via epithelial-mesenchymal transition (EMT) [7]. Factors released from CAFs into the tumor microenvironment play crucial roles in tumor growth, angiogenesis, metastasis, and resistance to therapy [5]. Thus, targeting CAFs directly to turn off their downstream effects or inhibiting CAF-secreted factors that stimulate tumor development and progression could represent a potential strategy for treating solid tumors.

Breast cancer is the most frequent cancer in women where an estimated 1.7 million women are diagnosed with breast cancer worldwide, every year [8]. Triple-negative breast cancer (TNBC) is a highly aggressive form of breast cancer, characterized by the absence of three receptors: estrogen (ER), progesterone (PR) and epidermal growth factor receptor 2 (HER2) [9]. TNBCs are resistant to most available targeted therapies because these treatments require the presence of at least one of these receptors to be effective in killing the cancer cells [9]. In addition, ~10–20% of all invasive breast cancers are TNBC [10]. TNBC metastasizes earlier and more frequently than other types of breast tumors; the 5-year survival rate for patients with TNBC is ~77% compared to 93% for all other breast cancer types [11]. Therefore, the development of novel therapeutic strategies for treating TNBC is urgently needed. Understanding the crosstalk between CAF subtypes and other cells in the TNBC microenvironment could open potential new avenues for cancer treatment.

Cancer-associated fibroblasts (CAFs) also coexist as heterogeneous populations, and several CAF subtypes with distinct molecular profiles have been identified in various cancers [6,12,13,14,15,16]. Although CAFs are the most prominent stromal components in solid tumors, identifying all possible subtypes and their specialized functions is far from complete. It has been previously shown that CAFs express high levels of alpha-smooth muscle actin (α-SMA/*Acta2*), CD90 (*Thy1*), platelet-derived growth factor receptors α or β (*Pdgfra/b*), integrin β1/CD29, podoplanin (*Pdpn*), osteonectin (*Sparc*), fibroblast activation protein (*Fap*), fibroblast-specific protein 1 (*S100a4*), caveolin 1 (*Cav1*) and vimentin (*Vim)* [3,14]. Several recent studies have used these markers to identify and characterize CAFs in various cancers [14,17,18,19]. However, these markers are far from being all-encompassing or completely specific to these cell subtypes, preventing us from identifying subtle differences among CAF subtypes using conventional methods. Single-cell RNA sequencing (scRNA-seq) allows us to profile gene expression in individual cells in a tissue with complex architecture and provides a high-resolution window into transcriptional differences. In turn, these molecular differences may lead to a better understanding of the function of each specific cell [20]. Furthermore, scRNA-seq enables us to discover rare cell types that until now may have been overlooked by traditional methods [21]. Several studies have utilized scRNA-seq to investigate CAF heterogeneity in solid tumors including pancreatic, breast and colorectal cancer, advancing our understanding of CAF heterogeneity [3,15,16], but no study to date has compared CAF subpopulations in various tumor types and also to fibroblast subpopulations present in healthy, normal tissues.

In this study, we characterized the fibroblast heterogeneity in a mouse allograft model of TNBC. Syngeneic mammary fat pad tumors were generated by injecting 4T1 breast cancer cells into BALB/c mice. Palpable tumors were dissected, and gene expression was profiled at single-cell level. The scRNA-seq analysis identified six CAF subpopulations in 4T1 mammary fat pad tumors including: 1) a CAF subpopulation with elevated expression of α-smooth muscle actin (α-SMA) and other contractile proteins including *Tnc*, *Tagln* and *Myl9*; 2) a subpopulation enriched in *Ly6c1* and inflammatory cytokines *Cxcl12*, *Il6* and *Ccl2;* and 3) a CAF subpopulation expressing *Cd74* and other MHC class II proteins. Furthermore, we compared the CAF signatures of 4T1 tumors to those of pancreatic tumors from a genetically engineered mouse model (GEMM), the KPC mouse [22], and from subcutaneous allografts with a cell line (mT3) derived from the KPC mice [23], and of normal tissue resident fibroblasts to determine their similarities and differences. α-SMA-high CAFs, inflammatory CAFs and MHC class II-expressing CAFs were found in both breast and pancreatic tumors and shared highly similar transcriptional profiles. Interestingly, cells with inflammatory CAF profile and MHC class II-expressing CAF profile were also found endogenous to healthy breast/pancreas tissues, suggesting that these types of fibroblasts are not induced by the tumor microenvironment and may play important roles in tissue homeostasis.

## 2. Results

### 2.1. scRNA-seq Reveals Transcriptional Profiles of CAFs in Murine Mammary Tumors

scRNA-seq was conducted on viable cells isolated from BALB/c-derived 4T1 orthotopic tumors using the 10x Genomics Chromium platform (Figure 1A). Of cells sequenced, 6420 cells met our quality control metrics and were further analyzed to identify various cell types in the tumor. A graph-based clustering using Seurat [24] identified 12 cell clusters (Figure 1B). By cross-referencing genes differentially expressed in each cluster to previously published cell-type specific markers, we assigned each cluster to its putative cell-type identity (Figure 1B). Cells in clusters 0, 2, 3, 5, 7, 8 and 9 expressed CD45 (*Ptprc*) and several other markers of myeloid and lymphoid lineages, and were classified as immune cells (Figure 1B,C, Appendix A). Immune cells accounted for 66.4% of all sequenced cells. Clusters expressing high levels of *Epcam* (clusters 1 and 6) were identified as epithelial/cancer cells and accounted for ~24.5% of all cells (Figure 1B,C, Appendix A). Cells in cluster 4 had high levels of *Thy1*, *Pdpn* and *Pdgfra* [25] and were identified as CAFs (Figure 1B,C, Appendix A). This cluster included 535 cells and accounted for ~8% of all cells analyzed. Cells in cluster 10 expressed high levels of *Pecam1* and *Mcam* and were identified as endothelial cells (Figure 1B,C, Appendix A). We also identified a small population of pericytes (cluster 11) (Figure 1B). Interestingly, pericytes shared many markers with CAFs including *Thy1* and *Pdgfrb* but also had unique markers such as NG2 (*Cspg4*), *Mcam* and *Rgs5* [26,27] (Figure 1C, Appendix A).

Next, we compared the gene expression profile of CAFs (cluster 4) to other cell types in the tumor. Although we were only able to identify fewer CAFs in 4T1 mammary tumors than generally expected (~8% CAFs vs ~66% immune cells and ~24% cancer cells), potentially due to the limitations of the single cell isolation method, our analysis identified a large number of genes significantly enriched in CAFs compared to other cell types (Appendix A). CAFs have been shown to play a major role in extracellular matrix (ECM) synthesis and organization [28]. Consistent with these functions, 4T1-derived CAFs showed enrichment for collagens (*Col1a1*, *Col1a2*, *Col3a1*, *Col5a1-3*, *Col6a1-3* etc.), proteoglycans (*Dcn*, *Lum*, *Bgn*, *Prg4* etc.) and glycoproteins (*Postn*, *Dpt*, *Tnc*, *Fbln*, *Fbn1* etc.) (Figure 1D,E). Although CAFs expressed basement membrane collagens *Col4a1* and *Col4a2*, the highest expression for these two genes was detected in pericytes and endothelial cells. CAFs also expressed high levels of several enzymes involved in processing and assembly of collagen into fibrils including Adamts-2, prolyl-4-hydroxylases (*P4ha1-3*), lysyl hydroxylases (*Plod1-3*) and lysyl oxidases (*Lox*, *Loxl1-3*) [29] (Figure 1D). Several ECM catabolic enzymes including matrix metalloproteinase-2 (*Mmp2*), matrix metalloproteinase-3 (*Mmp3*) and cathepsin K (*Ctsk*) were also enriched in CAFs (Figure 1D). CAFs also showed enrichment for bone morphogenic protein 1 (*Bmp1*), Tgf-β receptors (*Tgfbr2*, *Tgfrb3*), Wnt signaling pathway inhibitors (*Sfrp1*, *Sfrp2*, *Sfrp4*), complement pathway genes (*C1ra*, *C1s1*, *C3*, *C4b*), cytokines and cytokine receptors including *Cxcl1*, *Cxcl12*, *Cxcl14*, *Il1r1* and *Il11ra1*(Figure 1F, Appendix A). Cell adhesion proteins *Cdh11* and *Chl1* were also enriched in CAFs (Figure 1F).

### 2.2. Six Distinct Subtypes of CAFs Were Detected in Murine Mammary Tumors

To better understand CAF heterogeneity in syngeneic 4T1 mammary tumors, we performed scRNA-seq on an immune (CD45+) and cancer cell (Thy1.1+) depleted fraction of 4T1-Thy1.1 tumor single cell suspension (Figure 1A). Among the ~4000 cells sequenced, we detected epithelial (*Epcam*+), endothelial (*Pecam1*+), pericytes (*Mcam*+, *Rgs5*+) and low levels of immune cells (CD45+); these cells were excluded from subsequent analysis (Appendix A). The remaining ~1600 cells were classified as CAFs based on the expression of commonly used CAF markers (Appendix A). Unsupervised clustering of these CAFs identified six distinct clusters with unique gene expression signatures (Figure 2A and Appendix A). CAF markers *Pdpn*, *Thy1* and *Pdgfra* [25] and CAF-secreted collagens *Col1a1* and *Col3a1* were expressed in all these clusters (Figure 2B). Cells in cluster 0 expressed high levels of lymphocyte antigen 6 complex, locus C1 (*Ly6c1*) while cells in cluster 1 were highly enriched for alpha smooth muscle actin (α-SMA/*Acta2*) (Figure 2C); as such, these genes were chosen to represent these clusters. Cells in cluster 2 expressed high levels of cyclin-dependent kinase 1 (*Cdk1*) and other cell cycle genes including *Cenpa* and *Cenpf* and were identified as ‘dividing cells’ (Figure 2C). Leukocyte surface antigen *Cd53* was highly enriched in cluster 3 and was used as a marker for this cluster (Figure 2C). Cells in cluster 4 showed significant enrichment for cellular retinoic acid-binding protein 1 (*Crabp1*), while cells in cluster 5 showed enrichment for *Cd74* (Figure 2C). *Crabp1* and *Cd74* were chosen as representative markers of clusters 4 and 5, respectively. We also found that these clusters had distinct gene expression profiles with a significant number of genes differentially expressed between these six clusters (Figure 2D, Appendix A).

To explore the relationship among these CAF subtypes, we constructed a transcriptional trajectory of these cells on a pseudotime scale using Monocle [30]. Cells in cluster 0 (Ly6c1^high^) were distributed at one end of the pseudotemporal trajectory whereas cells in cluster 1 (α-SMA/Acta2^high^) resided at the other end, suggesting that these clusters are most divergent from each other (Figure 2E,F). Pseudotime analysis also suggested that dividing/cycling cells from cluster 2 (Cdk1^high^) included cells diverging from both Ly6c1^high^ and Acta2^high^ clusters (Figure 2E,F). Cluster 3 (Cd53^high^) cells existed along the trajectory. Cluster 4 (Crabp1^high^) resided next to Ly6c1^high^ cluster while cluster 5 (Cd74^high^) cells were mainly distributed at the other end of the trajectory, closer to the α-SMA^high^ cluster (Figure 2E,F).

The Ly6c1^high^ (cluster 0) and α-SMA^high^ (cluster 1) clusters together constituted ~79% of fibroblasts in the 4T1 tumor (Figure 2A) and were further analyzed to gain insights into their potential roles in tumor development and progression (Figure 3A,B). Ly6c1 is an antigen present in neutrophils, monocytes, dendritic cells, and T cells [31] and its function in fibroblasts is not yet known. The Ly6c1^high^ cluster also expressed other marker genes including ECM protein dermatopontin (*Dpt*), plasminogen-binding C-type lectin tetranectin (*Clec3b*), and hyaluronan synthase 1 (*Has1*), an enzyme responsible for cellular hyaluronan synthesis [32] at higher levels (Figure 3A). In addition, Ly6c1^high^ cluster showed enrichment for stem cell antigen-1 (*Ly6a*/Sca-1), serum amyloid A3 (*Saa3*), collagen 14a1 (*Col14a1*) which plays a regulatory role in collagen fibrillogenesis [33], a small leucine-rich proteoglycan osteoglycin (*Ogn*), proteoglycan 4 (*Prg4*), prolargin (*Prelp*), EGF-containing fibulin-like extracellular matrix protein 1 (*Efemp1*), and HtrA serine peptidase 3 (*Htra3*) (Appendix A). Hyaluronan, a major non-protein glycosaminoglycan component of the ECM, has been shown to promote cancer cell proliferation, migration, invasion, adhesion, EMT and cancer stem cell activation [34]. Increased *Has1* expression may contribute to increased hyaluronan synthesis in Ly6c1^high^ CAFs. Ogn plays a restrictive role in cancer progression [35,36]. Htra3 may also have a tumor-suppressive function [37]. Efemp1 and Prg4 can have either tumor-promoting or tumor-suppressive function [38,39,40,41]. We also observed elevated expression of transcripts encoding immune modulatory cytokines including interleukin-6 (*Il6*), interleukin 33 (*Il33*), chemokine (C-X-C motif) ligand 1 (*Cxcl1*), C-X-C motif chemokine 12 (*Cxcl12*), monocyte chemoattractant protein-1 (MCP-1/*Ccl2*) and monocyte-chemotactic protein 3 (MCP3/*Ccl7*) and several members of compliment pathway including *C3*, *C4b*, *C1s1* and *C1s2* in Ly6c1^high^ cluster (Figure 3C,D), suggesting that Ly6c1^high^ CAFs play a role in regulating the immune responses in the tumor microenvironment.

The α-SMA^high^ cluster (cluster 1), had a molecular signature similar to the myofibroblasts, which have been shown to play crucial roles in wound healing and pathological tissue remodeling [42,43]. Genes highly expressed in the α-SMA^high^ cluster but absent or significantly reduced in other clusters included: contractile proteins tropomyosins 1 and 2 (*Tpm1*, *Tpm2*) and myosin light chain 9 (*Myl9*), transgelin (*Tagln*), calponins (*Cnn2* and *Cnn3*), insulin-like growth factor-binding protein 3 (*Igfbp3*), tenascin C (*Tnc*) and transmembrane Protein 119 (*Tmem119*) (Figure 3B and Appendix A). Interestingly, the α-SMA^high^ cluster showed enrichment for several growth factor genes including transforming growth factor β (*Tgfb1* and *Tgfb2*), connective tissue growth factor (CCN2/*Ctgf*), placental growth factor (*Pgf*), vascular endothelial growth factor A (*Vegfa*) and *Wnt5a* (Figure 3D). These growth factors have been implicated in various aspects of cancer development and progression including cell proliferation, migration, invasion, EMT and angiogenesis, suggesting that α-SMA^high^ CAFs may promote tumor growth and progression [44,45,46,47,48]. We also observed that Ly6c1^high^ CAFs expressed higher levels of platelet derived growth factor receptor alpha (*Pdgfra*), a commonly used CAF marker, whereas platelet derived growth factor receptor beta (*Pdgfrb*) expression was higher in cells from the α-SMA^high^ cluster (Appendix A).

Although small in size, Cd53^high^ (cluster 3), Crabp1^high^ (cluster 4) and Cd74^high^ (cluster 5) clusters also displayed unique gene expression profiles, suggesting that these CAFs may also have distinct functions in the tumor microenvironment (Figure 2A,D). Cells in Cd53^high^ cluster showed significant transcriptional enrichment for desmin (*Des*), a cytoplasmic intermediate filament protein which plays a crucial role in structural integrity and function of muscle [49] (Appendix A). Other genes enriched in this cluster included matrix glycoprotein fibronectin 1 (*Fn1*) which has a well-established role in tumor development and progression [50,51,52], integrin alpha 1 (*Itga1*), syndecan 1 (*Sdc1*), matrix metalloproteinase inhibitors *Timp1*, *Timp2* and *Timp3* and galectin 3 (*Lgals3*), a potential regulator of cell migration, proliferation, angiogenesis, EMT and apoptosis [53] (Appendix A). Basement membrane collagens *Col4a1*, *Col4a2*, *Col18a1*, laminin A2 (*Lama2*) and *Mmp19*, a protease which has been reported to degrade several basement membrane proteins [54] were among the genes highly enriched in the Crabp1^high^ cluster (Figure 3E). This cluster also showed enrichment for perlecan (*Hspg2*), a major component of basement membranes, ECM proteins lumican (*Lum*), decorin (*Dcn*) and spondin 1 (*Spon1*), and insulin-like growth factor 1 (*Igf1*) (Appendix A). Cells in the Cd74^high^ cluster uniquely expressed high levels of MHC class II genes (*H2-Aa*, *H2-Ab1*, *H2-Eb*, *Cd74* etc.) which are normally expressed by antigen-presenting cells (Figure 3F). This cluster also expressed commonly used CAF markers such as *Pdpn*, *Pdgfra*, *Thy1*, *Col1a1*, *Col3a1* and *Dcn* at comparable levels to other CAF clusters confirming that these Cd74^high^ cells are CAFs and not immune cells (Appendix A). Expression of the pan-CAF markers such as *Pdpn*, *Pdgfra* and *Dcn* were extremely low in immune cells and epithelial cells whereas MHC class II genes were expressed in antigen-presenting immune cells at high levels (Appendix A). We also detected enrichment for transcripts encoding fibroblast-specific protein 1 (FSP1/*S100a4*), keratins *Krt7*, *Krt8*, *Krt14* and *Krt18* and claudins *Cldn3*, *Cldn4* and *Cldn7* in this cluster which were also enriched in epithelial/cancer cells (Figure 3F and Appendix A).

### 2.3. TNBC-Derived CAF Subtypes Share Molecular Features with Pancreatic Ductal Adenocarcinoma (PDAC)-Derived CAF Subtypes

Elyada et al. have previously demonstrated that three distinct CAF subtypes exist in PDAC: 1) “myofibroblastic CAFs” (myCAFs), that express high levels of α-SMA and other contractile genes such as *Tagln*, *Myl9* and tropomyosins; 2) “inflammatory CAFs” (iCAFs) which express low levels of α-SMA but high levels of cytokines and other markers such as *Ly6c1*, *Clec3b*, *Dpt* and *Has1*; 3) antigen-presenting CAFs (apCAFs) which express MHC class II-related genes and induce T cell receptor (TCR) ligation in CD4+ T cells in an antigen-dependent manner [15]. Here, we compared the profiles of breast and pancreatic tumor-derived CAFs to gain more insights into CAF heterogeneity across different solid tumor types.

We reproduced Elyada et al.’s findings using publicly available scRNA-seq data derived from a genetic mouse model of PDAC, the KPC mouse (Kras^+/LSL-G12D^; Trp53^+/LSL-R172H^; Pdx-Cre) (GSE129455) [15], and identified the three CAF subtypes (Figure 4A). All of these CAF subtypes expressed pan-CAF markers including *Thy1*, *Pdpn*, *Pdgfra* and *Col1a1* (Figure 4B). The CAF subtypes identified in PDAC included a Ly6c1^high^ cluster (cluster 0) which was identified as iCAFs based on the expression of markers including *Ly6c1*, *Clec3b*, *Has1*, *Dpt* and *Col14a1* and a α-SMA^high^ cluster (cluster 2) with high expression of *Acta2*, *Tagln*, *Myl9*, *Igfbp3* and *Tnc* which was identified as myCAFs (Figure 4A–C). As in the case for TNBC, PDAC-derived Ly6c1^high^ CAFs (iCAFs) also expressed genes coding for chemokines and other inflammatory mediators such as *Il6*, *Il33*, *Cxcl1*, *Cxcl12* and *Ccl7* while α-SMA^high^ CAFs (myCAFs) showed enrichment for growth factors *Tgfb1*, *Tgfb2* and *Ctgf* (Figure 3C,D and Figure 4C). We also identified a Cd74^high^ cluster (cluster 1) which was enriched for MHC class II-related genes *Cd74*, *H2-Aa*, *H2-Ab1* and *H2-Eb1* and was identified as apCAFs (Figure 4A,D). Interestingly, in both cancer types Cd74^high^ CAFs (apCAFs) showed enrichment for keratins including *Krt8* and *Krt18* and Fsp1 (*S100a4*), a fibroblast marker (Figure 3F and Figure 4D). A comparative analysis of 4T1- and KPC-derived CAFs further confirmed that iCAFs, myCAFs and apCAFs are highly similar in both breast and pancreatic cancers (Appendix A).

Additionally, we generated syngeneic tumors in immunocompetent C57BL/6 mice by injecting the KPC-derived PDAC cell line mT3, subcutaneously (SQ) [23]. Of the ~2500 cells sequenced from an immune-depleted tumor, 434 cells were classified as CAFs, forming two distinct clusters (Figure 4E). Both CAF clusters expressed pan-CAF markers *Pdpn*, *Thy1*, *Pdgfra* and *Col1a1* (Figure 4F). Further analysis identified these 2 CAF subtypes as: 1) Ly6c1^high^ CAFs (iCAFs) based on enrichment for *Ly6c1*, *Clec3b*, *Has1* and *Dpt* and 2) α-SMA^high^ CAFs (myCAFs) based on enrichment for *Acta2*, *Tagln*, *Tnc* and *Myl9* (Figure 4G). Interestingly, we did not detect any Cd74^high^ cells in the syngeneic mT3 SQ tumors.

Regardless of the tumor type or tumor site, Ly6c1^high^ CAFs (iCAFs) showed enrichment for *Il33*, *Il6*, *Ccl7*, *Cxcl1* and *Cxcl12* whereas α-SMA^high^ CAFs (myCAFs) expressed high levels of *Tgfb1*, *Tgfb2* and *Ctgf* (Figure 3D and Figure 4C,G). This comparative gene expression analysis showed high concordance between Ly6c1^high^, α-SMA^high^ and Cd74^high^ CAFs in both breast and pancreatic cancers, but points out that SQ tumor models have some limitations.

### 2.4. Cells with Ly6c1^high^ and Cd74^high^ CAF Profiles are Present in Normal, Healthy Tissues

To understand the molecular profiles of tissue resident fibroblasts, we isolated Pdgfra-expressing fibroblasts from the mammary fat pads of normal BALB/c mice [55] and performed single-cell sequencing on these cells. The ~1600 sequenced cells clustered in 3 distinct subtypes (Figure 5A). All clusters expressed CAF/fibroblast markers *Pdgfra*, *Dcn*, *Postn* and *Col1a1*; however, *Pdpn* and *Thy1* expression was more restricted to clusters 0 and 2 (Figure 5B,C). Interestingly, clusters 0 and 2 were enriched for markers of Ly6c1^high^ CAFs including *Ly6c1*, *Clec3b*, *Dpt*, *Has1* and *Col14a1*, and cytokines including *Ccl2*, *Ccl7*, *Il6*, *Cxcl1* and *Cxcl12,* although there were some differences in the expression of these genes between these two clusters (Figure 5C,D, Appendix A). This suggests that fibroblasts with a Ly6c1^high^ CAF profile exist in normal tissues as well. Clusters 0 and 2 fibroblasts were also enriched for ECM remodeling enzymes including *Mmp2*, *Ctsk*, *Adamts5* and *Htra3* but showed some differences in the expression of core ECM genes (Figure 5E and Appendix A, Appendix A). We did not detect any α-SMA^high^ or Cd74^high^ clusters among normal mammary fibroblasts although an extremely small fraction of cells expressed these genes (Figure 5D). However, none of these cells expressed *Cd53* or *Crabp1*. Cluster 1 showed enrichment for several long non-coding RNAs including maternally expressed 3 (*Meg3)*, nuclear paraspeckle assembly transcript 1 (*Neat1)*, X-inactive specific transcript (*Xist)* and metastasis associated lung adenocarcinoma transcript 1 (*Malat1)* and the expression of Ly6c1^high^ CAF markers were extremely low in this cluster (Appendix A and Figure 5C).

A comparative analysis of fibroblasts derived from normal mammary fat pad and 4T1 tumor-derived CAFs revealed that Ly6c1^high^ CAFs from the tumor share high molecular similarity with Ly6c1^high^ fibroblasts from normal mammary fat pad (Appendix A), expressing common markers such as *Ly6c1*, *Clec3b*, *Dpt*, *Ly6a*, *Htra3*, *Has1* and *Col14a1* (Appendix A). However, transcripts encoding cytokines *Cxcl12* and *Il33* had a significantly higher expression in the tumor-derived CAFs, suggesting a tumor microenvironment induced activation of these genes in this subpopulation (Appendix A). Expression of markers of other CAF subtypes were mainly restricted to the tumor (Appendix A).

We also analyzed a publicly available normal pancreas scRNA-seq data (GEO: GSM3577882) [56] to determine fibroblast diversity in this tissue. Fibroblasts were identified based on the expression of markers *Pdgfra*, *Pdpn*, *Thy1*, *Dcn*, *Col1a1* and *Col3a1* and were further analyzed to identify distinct subtypes. In normal pancreas, we identified 3 clusters expressing fibroblast markers (Figure 5F,G). Cluster 0 showed significant enrichment for Ly6c1^high^ inflammatory CAF markers *Ly6c1*, *Clec3b*, *Has1*, *Dpt* and *Col14a1* (Figure 5H,I). Cluster 1 also expressed these markers at some level, and cells in both cluster 0 and 1 transcriptionally expressed cytokines including *Ccl2*, *Ccl7*, *Il6*, *Cxcl1* and *Cxcl12* (Figure 5I). Cluster 1 also showed enrichment for long noncoding RNAs *Meg3*, *Neat1*, *Xist* and *Malat1* that were found to be enriched in a subset of normal mammary fibroblasts (Appendix A). Interestingly, cluster 2 showed enrichment for *Cd74* and other genes enriched in Cd74^high^ CAFs including *H2-Ab1*, *H2-Aa*, *Krt8*, *Krt18* and fibroblast marker *Fsp1* (S100a4), suggesting that cells with Cd74^high^ CAF features are also present in the normal pancreas (Figure 5J).

Using flow cytometry, we confirmed that Ly6c1^high^ fibroblasts are present in both mammary tumors and naïve mammary fat pad. Fibroblasts were isolated by removing immune (anti-CD45) and cancer cells (anti-CD90.1/Thy1.1) and then selecting for cells expressing CAF marker Thy1 (CD90.2/Thy1.2) (Figure 2B). About 2.4% of cells in the tumor and ~30% of cells in the normal mammary fat pad expressed CD90.2/Thy1.2 (Figure 6A–F). About 40% of the cells in this fibroblast-enriched fraction expressed Ly6c in both the tumor and naïve mammary fat pad. Interestingly, a small proportion of these Thy1+ cells expressed MHC class II protein (Figure 6G–I), confirming that cells with Cd74^high^ CAF profile are also present in both tumor and normal mammary tissue. Tumor tissue had a significantly higher proportion of these MHC class II-expressing cells than normal mammary fat pad (Figure 6G–I).

## 3. Discussion

In this study, we have carried out scRNA-seq of 4T1 mammary and mT3 pancreatic syngeneic tumors to investigate CAF heterogeneity across TNBC and PDAC tumor types. Our study identified six CAF subtypes in TNBC with distinct gene expression profiles: 1) Ly6c1^high^ CAFs; 2) α-SMA^high^ CAFs; 3) dividing/cycling CAFs; 4) Cd53^high^ CAFs; 5) Crabp1^high^ CAFs and 6) Cd74^high^ CAFs. A comparison of this data to data from KPC mice-derived pancreatic tumors revealed that 3 of these CAF populations, Ly6c1^high^ CAFs (iCAFs), α-SMA^high^ CAFs (myCAFs) and Cd74^high^ CAFs (apCAFs) exist in both tumor types. These 3 CAF subtypes have also been detected in human PDACs [15].

A high proportion of cells in both breast and pancreatic cancers expressed *α-SMA* (*Acta2*). While the existence of myofibroblastic (α-SMA^high^) CAFs in solid tumors are well-established [3,57], the number of *α-SMA*-expressing fibroblasts was extremely low in normal breast and pancreatic tissue suggesting that this subtype emerges during tumorigenesis. In 4T1 mammary fat pad tumors, α-SMA^high^ CAFs showed enrichment for several growth factor transcripts including *Tgfb1*, *Tgfb2*, *Ctgf*, *Pgf*, *Vegfa* and *Wnt5a* (Figure 2C,D). TGF-β is a multifunctional cytokine with a well-established role in fibroblast to myofibroblast differentiation, EMT and immune regulation [44,58,59,60]. The protein Vegfa is a key regulator of both physiological and pathological angiogenesis [46]. Other factors secreted by α-SMA^high^ CAFs such as Ctgf, Pgf and Wnt5a have been implicated in cancer cell proliferation, migration, invasion, EMT and/or angiogenesis [45,47,48], suggesting that α-SMA^high^ CAFs may promote tumor development and progression. Interestingly, a previous study has shown that instead of eradicating the tumor, depletion of myofibroblasts in PDAC caused an increase in tumor invasion; an event associated with decreased survival. These findings suggest that α-SMA^high^ CAFs may also function to keep the tumor in check [61].

The Ly6c1^high^ CAFs expressed high levels of transcripts encoding inflammatory cytokines including *Ccl2*, *Il6*, *Il33* and *Cxcl12*. Stromal fibroblast derived Ccl2 has been shown to promote tumor progression and contribute to immune evasion [62,63]. IL6 has been shown to drive EMT, metastasis and therapy resistance in cancer [64,65,66,67]. IL6 is also plays a role in the generation of tumor-associated macrophages by skewing monocyte differentiation into tumor-associated macrophage [68]. It has been shown that Il33 plays a role in tumor growth, metastasis, neo-angiogenesis, and evading programmed cell death [69]. Multiple studies have shown that Cxcl12 promotes tumor angiogenesis, tumor cell proliferation and chemoresistance [70]. Recently, Costa et al. analyzed the expression of six previously known CAF markers (FAP, integrin β1/CD29, α-SMA, S100-A4/FSP1, PDGFRβ, and CAV1) in human breast cancer and discovered four different CAF subpopulations that expressed these markers at varying levels, including a subtype which promoted immunosuppression through a Cxcl12-dependent mechanism [14]. These studies together suggest that Ly6c1^high^ CAFs may play a role in immune suppression in the tumor microenvironment. Interestingly, we found fibroblasts with Ly6c1^high^ CAF profile in normal breast and pancreatic tissues, suggesting that Ly6c1^high^ CAFs might have originated from resident fibroblasts. Alternatively, the resident fibroblasts with high *Ly6c1* expression got recruited into the tumor during tumor development. It has been shown that IL1 signaling through IL1R promotes the Ly6c1^high^ CAF (iCAF) phenotype via JAK–STAT signaling and inhibition of JAK/STAT signaling shifts iCAFs to a myofibroblastic phenotype in PDAC [59]. Therefore, Ly6c1^high^ fibroblasts present in normal tissue may differentiate into α-SMA^high^ CAFs (myCAFs) during tumor progression. Further studies are required to understand the specific cues from the tumor microenvironment that drive these phenotypes.

The Cd74^high^ CAFs (apCAFs) expressed CD74 and other MHC class II proteins which are normally expressed by antigen presenting cells in the immune system. In addition, Cd74^high^ CAFs showed enrichment for transcripts encoding keratins including *Krt7*, *Krt8* and *Krt18* and Fsp1 (*S100a4*), a protein predominately expressed in fibroblasts. Elyada et al. have shown the presence of apCAFs in both mouse and human PDAC samples [15]. They also showed that apCAFs can present antigens to CD4+ T cells [15]. Another scRNA-seq study showed the presence of a cell population with ‘apCAF’ profile in normal pancreas and identified it as mesothelial cells [71]. In normal pancreas, these cells expressed *Pdpn* but, lacked *Pdgfra* expression [71]. Here we showed that cells with apCAF profile are present in both normal mammary tissue and mammary tumors. In breast/pancreatic tumors and normal breast/pancreatic tissues, Cd74^high^ cells expressed fibroblast marker *Fsp1/S100a4* (Figure 3F, Figure 4D and Figure 5J) whereas *Pdgfra* expression appeared to be restricted to tumor tissue only (Appendix A, Figure 4B and Figure 5G). Future labeling and lineage tracing experiments will be able to conclusively determine the origin of these cells. It has been hypothesized that apCAFs might have an immune modulatory role in the tumor microenvironment [15], and the possibility exists that these cells also protect healthy tissue from auto-immune reactions in conditions such as lupus or pancreatitis.

Recently, using scRNA-seq, Bartoschek et al. identified four transcriptionally distinct subpopulations of CAFs known as vascular CAFs (vCAFs), matrix CAFs (mCAFs), developmental CAFs (dCAFs) and cycling CAFs (cCAFs) in the genetically engineered MMTV-PyMT mouse model of breast cancer [16]. The vCAFs showed enrichment for several genes involved in vascular development including *Mcam* and *Rgs5* and likely have a perivascular origin [16]. Upon re-analysis of this data we detected enrichment for myofibroblast markers *Acta2*, *Tagln*, *Myl9* and *Tpm2* in vCAFs, suggesting that some of these cells acquired a myofibroblastic molecular profile (Appendix A). It was suggested that mCAFs originate from resident fibroblasts, and the proportion of mCAFs decrease with tumor progression [16]. We found that mCAFs were enriched for *Crabp1* and several other markers of Crabp1^high^ CAFs from 4T1 tumors including *Lama2*, *Spon1*, *Dcn*, *Lum* and *Mmp19*. We also detected weak *Ly6c1* expression in MMTV-PyMT-derived mCAFs raising the possibility that mCAFs (Crabp1^high^) might have originated from Ly6c1^high^ fibroblasts (Appendix A). In agreement with this, Crabp1^high^ CAFs resided adjacent to Ly6c1^high^ CAFs on the pseudotemporal trajectory (Figure 2E). Further studies are required to understand specific functions of Crabp1^high^ CAFs in the tumor microenvironment. However, we did not detect any Cd74^high^ CAF clusters in this dataset. Cd53^high^ CAFs were also not identified in this dataset or in the KPC model of pancreatic cancer. However, we could identify Cd53^high^ CAFs in our initial 4T1 scRNA-seq data (Appendix A). Future studies will determine whether Cd53^high^ CAFs are a subtype with distinct origin and function or merely represent a transitional state during the differentiation of CAFs.

To our knowledge this is the first report that comprehensively compares CAF heterogeneity in multiple tumor types in parallel to the healthy, tissue of origin, using a single cell RNA sequencing approach. We have identified six CAF subtypes in TNBC with unique transcriptomic profiles, adding several subtypes to some that have been widely described in the literature. Our findings, in line with numerous other studies, suggest that specific cues from the microenvironment are necessary to maintain these unique molecular features; many of these subtypes have been shown to acquire a myofibroblasts-like phenotype when cultured in 2D or 3D [15,72]. A major limitation of our study is that we only examined tumors at one timepoint. It is possible that these populations shift as the tumor grows in size, as it acquires resistance to therapy, or as it metastasizes [16]. Future studies should aim to compare multiple tumor types and stages to further refine our understanding of CAF heterogeneity in solid tumors as well as to begin to elucidate individual contributions to tumor function.

## 4. Materials and Methods

### 4.1. Cell Lines

4T1 cells were purchased from ATCC (Manassas, VA, USA). The 4T1-Thy1.1 cell line was graciously provided as a gift from Dr. Julian Lum [73]. 4T1-Thy1.1 cells express a non-native isoform of Thy1, Thy1.1 (CD90.1) which differs from native isoform Thy1.2 (CD90.2) in one amino acid. The mT3 cell line was developed from organoids isolated from KPC mouse PDAC lesions [23]. The cell line is syngeneic and forms tumors in immune-competent C57BL/6 mice.

### 4.2. Generation of Orthotopic Mammary Tumors and Tumor Digestion

10-week-old female BALB/c mice (Jackson Laboratories, Bar Harbor, ME, USA) were injected with 25,000 4T1 cells (ATCC, Manassas, VA, USA) or 4T1-Thy1.1 cells in a 1:1 suspension of Matrigel (Corning, Corning, NY, USA; Catalog no. 354234) and phosphate buffered saline (PBS) into mammary fat pad (MFP) to establish tumors. Mice were euthanized 19–26 days post injection and the tumors were dissected from these mice. Single cell suspensions were generated by passing the tumor through a syringe without a needle followed by a 1h digest with shaking at 37 °C in 100 μg/mL DNase I (Roche, Basel, Switzerland; catalog no. 11284932001), 300 U/mL collagenase/100U/mL hyaluronidase (STEMCELL Technologies, Vancouver, BC, Canada; catalog no. 07912), 0.6 U/mL Dispase II (Roche, Basel, Switzerland; catalog no. 4942078001) in DMEM/D12 with 10% fetal bovine serum (FBS) (ThermoFisher, Waltham, MA, USA). Digests were filtered through a 100 µm cell strainer prior to debris removal (Miltenyi Biotec, Bergisch Gladbach, Germany; catalog no. 130-109-398) and resuspended in BD FACS Pre-Sort Buffer (BD, Franklin Lakes, NJ, USA; catalog no. 563503) followed by red blood cell lysis (ACK lysis buffer, ThermoFisher, Waltham, MA, USA; catalog no. A1049201) prior to downstream applications. All animal experimental procedures were completed under an approved institutional animal care and use committee (IACUC) protocol at Lawrence Livermore National Laboratory and conforming to the National Institutes of Health Guide for the care and use of laboratory animals.

### 4.3. Single Cell Sequencing of 4T1 Tumors and Data Analysis

4T1 tumor from a female BALB/c mouse was processed and cell suspension was prepared as described above. Two subsequent washes in sterile PBS + 0.04% non-acetylated bovine serum albumin were performed to further remove debris from final suspension. Cell pellets were resuspended in PBS with 0.04% non-acetylated BSA prior to single cell sequencing preparation using Chromium Single Cell 3′ GEM, Library & Gel Bead Kit v3 (10x Genomics, Pleasanton, CA, USA; Catalog no. 1000075) on a 10 Genomics Chromium Controller following manufacturers protocol. scRNA-Seq libraries were sequenced using Illumina (San Diego, CA, USA) NextSeq 500.

The Cell Ranger Single-Cell Software Suite (10x Genomics, Pleasanton, CA, USA) was used to perform sample demultiplexing, barcode processing, and single-cell 3′gene counting. Samples were aligned to the mouse genome (mm10) using “cellranger mkfastq” with default parameters. Unique molecular identifier (UMI) counts were generated using “cellranger count”. Further analysis was performed in R using the Seurat package [74]. First, cells with fewer than 500 detected genes per cell and genes that were expressed by fewer than 5 cells were filtered out. To remove noise from droplets containing more than one cell, cells with more 7800 measured genes were filtered out. Dead cells were excluded by retaining cells with less than 5% mitochondrial reads. After removing all the unwanted cells from the dataset, we normalized the data by employing a global-scaling normalization method “LogNormalize”. Subsequently, we identified the 2000 most variable genes in the dataset. The data was then scaled to a mean of 0 and variance of 1 and the dimensionality of the data was reduced by principal component analysis (PCA) using the previously determined 2000 variable genes. Subsequently, we constructed a K-nearest-neighbor (KNN) graph based on the Euclidean distance in PCA space using the “FindNeighbors” function (using dimensions 1 to 15) and applied Louvain algorithm to iteratively group cells together by “FindClusters” function (resolution = 0.3). A non-linear dimensional reduction was then performed via uniform manifold approximation and projection (UMAP) using the first 15 principle components. A total of 12 clusters were identified in the 4T1 scRNA-seq data. Raw count matrix is available online in Dryad data repository (https://datadryad.org, doi:10.6071/M3238R).

### 4.4. Single Cell Sequencing and Analysis of Stromal Cell-Enriched Fraction

4T1-Thy1.1 tumor from a female BALB/c mouse was processed as described above to obtain single cell suspensions. To enrich for stromal cells in the sample, immune cells and Thy1.1 expressing cancer cells were removed from the single cell suspension using magnetic cell separation with CD90.1 and CD45 MicroBeads (Miltenyi Biotec, Bergisch Gladbach, Germany; Catalog no. 130-121-273 and 130-052-301, respectively) depletion in combination with LS magnetic separation columns (Miltenyi Biotec, Bergisch Gladbach, Germany; Catalog no. 130-042-401). Cells were prepared for single cell sequencing using Chromium Single Cell 3′ GEM, Library & Gel Bead Kit v3 (10x Genomics, Pleasanton, CA, USA; Catalog no. 1000075) on a 10x Genomics Chromium Controller following manufacturers protocol and sequenced using Illumina (San Diego, CA, USA) NextSeq 500 at median sequencing depth of ~30,000 reads.

The scRNA-seq data was analyzed using Cell Ranger Single-Cell Software Suite (10x Genomics, Pleasanton, CA, USA) as described above to obtain UMI counts. Subsequent analysis was performed in Seurat [74] as described above and clusters of various cell types were identified. Clusters with high expression of *Ptprc* and *Epcam* were regarded as contaminants of immune cells and cancer cells, respectively and were discarded from subsequent analysis. Clusters with high expression of *Pecam1* and *Rgs5* were identified as endothelial cells and pericytes, respectively and were also discarded from further analysis. Resulting clusters showed enrichment for CAF markers *Pdpn*, *Pdgfra*, *Thy1, Col1a1* and *Dcn*. Raw expression values were obtained for cells in these clusters and were further analyzed using Seurat as described above. To remove noise from droplets containing more than one cell, cells with more 5800 measured genes were filtered out. Subsequently, after performing a log-normalization we identified the 2000 most variable genes in the dataset. The data was then scaled, and the dimensionality of the data was reduced by PCA. Clusters of cells were identified on the basis of a shared-nearest neighbor graph between cells and the Louvain algorithm as described above. Subsequently, a dimensional reduction was performed via UMAP as described above and various CAF subtypes were identified. We subsequently repeated this analysis by varying the number of variable genes used for dimensionality reduction to further confirm the presence of these CAF subtypes in 4T1 tumors. Raw count matrix is available online in Dryad data repository (https://datadryad.org, doi:10.6071/M3238R).

### 4.5. Single-Cell Sequencing of Syngeneic mT3 Tumors and Data Analysis

To generate mT3 tumor, 10-week-old female C57BL/6 mice were injected with 25,000 mT3 cells subcutaneously (SQ) into the back flank in a 1:1 suspension of Matrigel (Corning, Corning, NY, USA; Catalog no. 354234) and PBS. At 3 weeks post injection the tumor was dissected and processed as described above to obtain single cell suspensions. Subsequently, immune cells and blood cells were removed by CD45+ magnetic bead-based depletion (Miltenyi Biotech, Bergisch Gladbach, Germany; Catalog no. 130-052-301) and ACK lysis buffer (ACK lysis buffer, ThermoFisher, Waltham, MA, USA catalog no. A1049201), respectively, following manufacturer’s guidelines. Remaining cells were prepared for single cell sequencing using Chromium Single Cell 3ʹ GEM, Library & Gel Bead Kit v3 (10x Genomics, Pleasanton, CA, USA; catalog no. 1000075) on a 10x Genomics Chromium Controller following manufacturers protocol and sequenced using Illumina (San Diego, CA, USA) NextSeq 500. The scRNA-seq data was demultiplexed and processed using Cell Ranger Single-Cell Software Suite (10x Genomics, Pleasanton, CA, USA) as described above to obtain UMI counts. Subsequent analysis was performed in Seurat [74] as described above and clusters of various cell types including cancer cells and CAFs were identified. Clusters expressing CAF markers *Pdpn*, *Pdgfra*, *Thy1* and *Dcn* were extracted and further analyzed using Seurat as described above and various CAF subtypes were identified. Raw count matrix is available online in Dryad data repository (https://datadryad.org, doi:10.6071/M3238R).

### 4.6. Isolation and Sequencing of Fibroblasts from Normal Mammary Fat Pad

Fourth and fifth mammary fat pads were collected from four BALBc mice, aged approximately 12 weeks [55]. Tissue was minced and then digested in 50 mL RPMI with 10% heat inactivated (HI) FBS and Penicillin-Streptomycin and 1 mg/mL Collagenase IV (ThermoFisher, Waltham, MA, USA). After an hour, the suspension was strained using a 70 µm cell strainer and centrifuged at 500× *g* for 15 min. The supernatant was discarded, and the cells were resuspended in 1 mL ACK (ThermoFisher, Waltham, MA, USA) for five minutes. 10 mL of PBS with 1% HI FBS was added and the cells were spun down at 500× *g* for 5 min. The supernatant was discarded, and the cells were resuspended in 0.5 mL of PBS with 1% HI FBS. To identify fibroblasts, cells were stained with the following antibodies at a 1:100 dilution: anti-CD326 (FITC, clone G8.8, BioLegend, San Diego, CA, USA), anti-CD45 (APC-Cy7, clone 30-F11, BioLegend, San Diego, CA, USA), anti-CD140a (Super Bright 436, clone APA5, ThermoFisher, Waltham, MA, USA), and 7AAD to assess viability. Cells were then sorted on a BD Aria to gate for CD140a^+^ EPCAM^−^ CD45− 7AAD^−^ cells (fibroblasts). Cells were prepared for single cell sequencing using Chromium Single Cell 3ʹ GEM, Library & Gel Bead Kit v3 (10x Genomics, Pleasanton, CA, USA; catalog no. 1000075) on a 10x Genomics Chromium Controller following manufacturers protocol and sequenced using Illumina NextSeq 500.

The scRNA-seq data was demultiplexed and processed using Cell Ranger Single-Cell Software Suite (10x Genomics, Pleasanton, CA, USA) as described above to obtain UMI counts. Subsequent analysis was performed in Seurat [74]. To remove noise from droplets containing more than one cell, cells with more 5900 measured genes were filtered out. All the other filtering steps and data normalization were performed as described above for 4T1 tumors. Prior to PCA, we identified the 2000 most variable genes and PCA was performed in this gene space. Clusters of cells were identified on the basis of a shared-nearest neighbor graph between cells as described above. First 15 principal components were provided as an input for dimensionality reduction via UMAP. Raw count matrix is available online at in Dryad data repository (https://datadryad.org, doi:10.6071/M3238R).

Comparative analysis of normal mammary fibroblasts and 4T1tumor-derived CAFs was performed using Seurat. The data was pre-processed and normalized as described above and 2000 most variable features were identified. Subsequently, integration anchors were identified using ‘FindIntegrationAnchors’ function and both datasets were integrated to generate a new integrated dataset. The integrated data was scaled, dimensionality was reduced, and the data was further analyzed as described above to identify clusters of CAF-subtypes.

### 4.7. Analysis of Cancer-Associated Fibroblasts from Mouse Pancreatic Tumors

Fibroblast-enriched scRNA-seq data from PDAC tumors of 4 KPC mice were obtained from GEO (GSE129455) as log transformed gene-by-cell count matrix. Subsequent data analysis was carried out in R with Seurat [74] as described above and various cell types were identified. Clusters of cells expressing CAF markers *Pdgfra*, *Pdpn*, *Thy1* and *Col1a1* were further analyzed and various CAF subpopulations were identified. Comparative analysis of 4T1 tumor-derived and PDAC-derived CAFs were performed using Seurat as described above.

### 4.8. Analysis of Normal Fibroblasts from Mouse Pancreas

From GEO (GSE125588) we obtained processed scRNA-seq files (barcodes.tsv, genes.tsv and matrix.mtx generated by Cell Ranger) associated with normal pancreas. The data was analyzed using Seurat [74] as described above to identify various cell types. Clusters of cells expressing fibroblast markers *Pdgfra*, *Dcn* and *Col1a1* were extracted and further analyzed using Seurat to identify various fibroblast subpopulations.

### 4.9. Analysis of Mammary Tumors from MMTV-PyMT Mice

scRNA-seq data for fibroblasts isolated from mammary tumors of two MMTV-PyMT mice were obtained from GEO (GSE111229) as raw counts. The data was analyzed using Seurat as described above to identify various CAF subtypes [74]. vCAFs, mCAFs, cCAFs and dCAFs were identified based on previously established markers [16].

### 4.10. Trajectory Finding

Single-cell pseudotime trajectories of CAFs were constructed with Monocle [30]. Expression data, phenotype data, and feature data were extracted from the Seurat object and a Monocle “CellDataSet” object was constructed using the “newCellDataSet” function. Highly variable genes from Seurat object were used as ordering genes. Dimensionality reduction was performed using the DDRTree algorithm implemented Monocle via the “reduceDimension” function. Cells were ordered along the trajectory using the “orderCells” method with default parameters.

### 4.11. Flow Cytometric Analysis of Inflammatory and MHC Class II-Expressing Fibroblasts

Single cell suspensions of normal mammary fat pads (*n* = 6) or 4T1 tumor-bearing BALB/c fat pads (*n* = 5) were generated as described above. Antibody staining was performed by incubating the cells with 1:100 dilution of APC/Cyanine7 anti-mouse I-A/I-E (MHC class II), Brilliant Violet 421™ anti-mouse CD90.2, PerCP/Cyanine5.5 anti-rat CD90/mouse CD90.1 (Thy-1.1), PerCP/Cyanine5.5 anti-mouse CD45, FITC anti-mouse Ly-6C (Biolegend, San Diego, CA, USA; Catalog no. 107627, 105341, 202515, 103131, 128005, respectively) for 30 min prior to resuspension in PBS with 1% FBS and 7-AAD Viability Staining Solution (BioLegend, San Diego, CA, USA; Catalog No. 420403). Cells were then analyzed on a BD (San Jose, CA, USA) FACSMelody instrument. Viable fibroblasts populations were identified as 7AAD−/CD45−/CD90.1−/CD90.2+ cells. Data analysis was performed using FlowJo software (v10.6.2).

## 5. Conclusions

This study systematically examined CAF heterogeneity in TNBCs and uncovered several CAF subpopulations in mammary tumor microenvironment. A comparison of breast and pancreatic-tumor-derived CAFs revealed that myofibroblast-like CAFs, inflammatory CAFs and MHC class II-expressing CAFs share similar molecular profiles in both tumor types. Our study also suggests that inflammatory fibroblasts and MHC class II-expressing fibroblasts are endogenous to healthy tissues, suggesting that these fibroblast subtypes are not induced by the tumor microenvironment but, are likely recruited to the tumor during tumor development. Further studies are required determine the specific functions of these CAF subpopulations in tumor development and progression. Effectively targeting these CAF subtypes could prevent cancer progression.

## Figures and Tables

**Figure 1 cancers-12-01307-f001:**
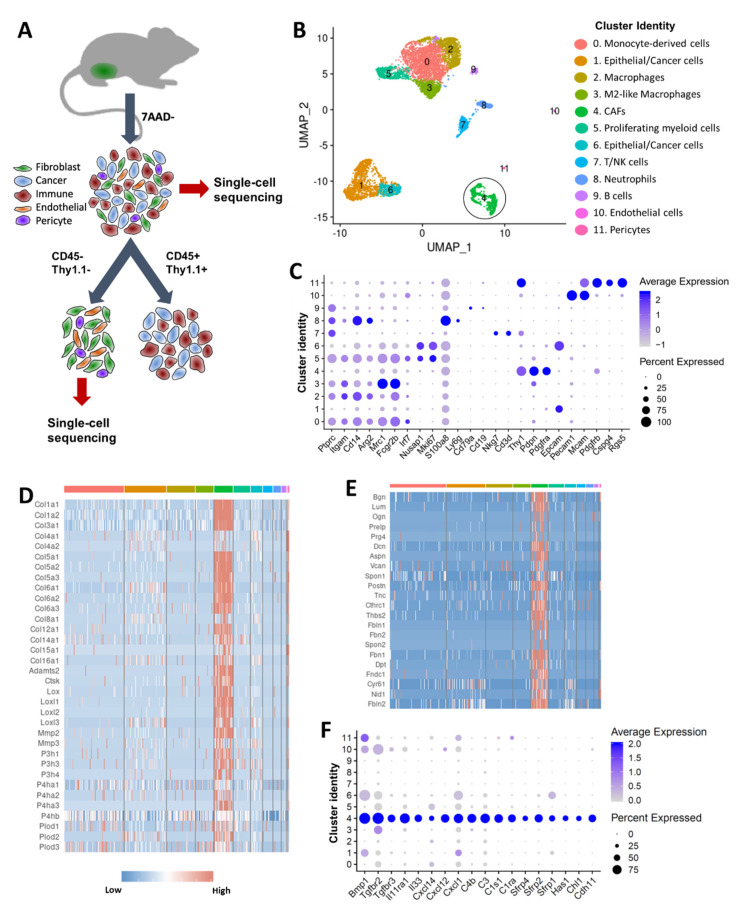
Single cell analysis of 4T1 mouse mammary tumors. (**A**) Graphical representation of the experimental workflow. 4T1 syngeneic tumors were dissociated into single cells, and two cell fractions were generated: (1) a viable cell fraction (7AAD−) and (2) immune-depleted stromal cell fraction (obtained by depleting CD45+ immune cells and Thy1.1+ cancer cells). Cells from both fractions were subjected to single cell sequencing using the 10x Genomics Chromium platform. (**B**) Cell clusters from 10x Genomics scRNA-seq analysis visualized by Uniform Manifold Approximation and Projection (UMAP). Colors indicate clusters of various cell types (CAFs in black circle). (**C**) Dot plot showing the expression of selected markers of various cell types. Dot size represents the fraction of cells expressing a specific marker in a particular cluster and intensity of color indicates the average expression level in that cluster. (**D**) Heatmap showing high levels of collagens and collagen-processing enzymes in CAFs. (**E**) Heatmap showing high levels of key proteoglycans and glycoproteins in CAFs. (**F**) Dot plot showing the expression of a subset of genes enriched in CAFs. Dot size represents the fraction of cells expressing a specific marker in a particular cluster and intensity of color indicates the average expression in that cluster.

**Figure 2 cancers-12-01307-f002:**
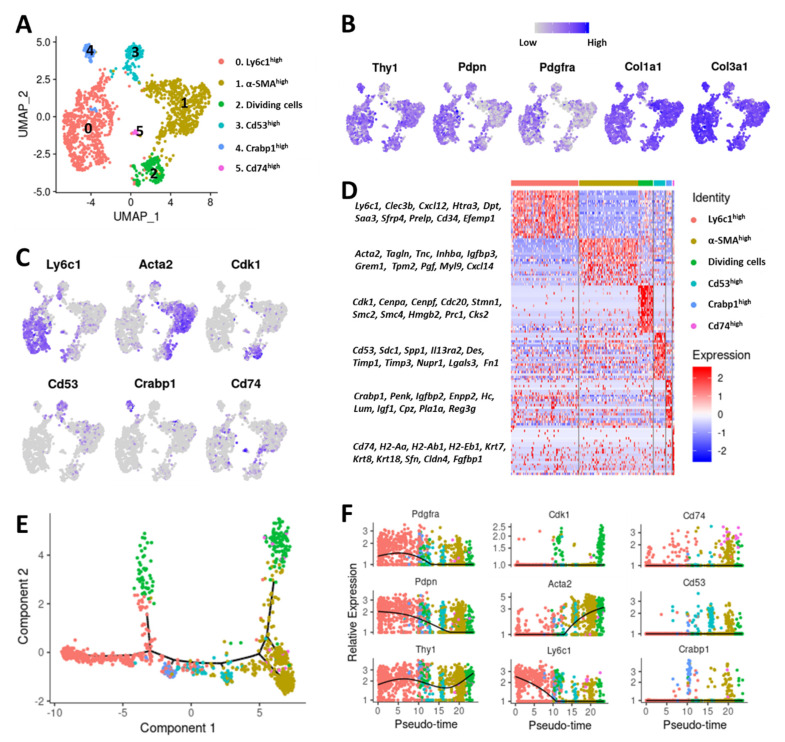
Characterization of CAF subtypes in 4T1 breast cancer. (**A**) Cell clusters from 10x Genomics scRNA-seq analysis visualized by UMAP. Colors indicate various CAF subtypes. (**B**) Feature plots showing the expression of commonly used CAF markers in various CAF subtypes. Legend shows a color gradient of normalized expression. (**C**) Markers of various CAF subtypes that were used to denote each subtype. (**D**) Heatmap showing a subset of genes differentially expressed between the 6 CAF subtypes. (**E**) Monocle pseudospace trajectory colored based on CAF clusters in (**A**). (**F**) Expression of CAF markers on a pseudotime scale (colored based on CAF clusters in (**A**)).

**Figure 3 cancers-12-01307-f003:**
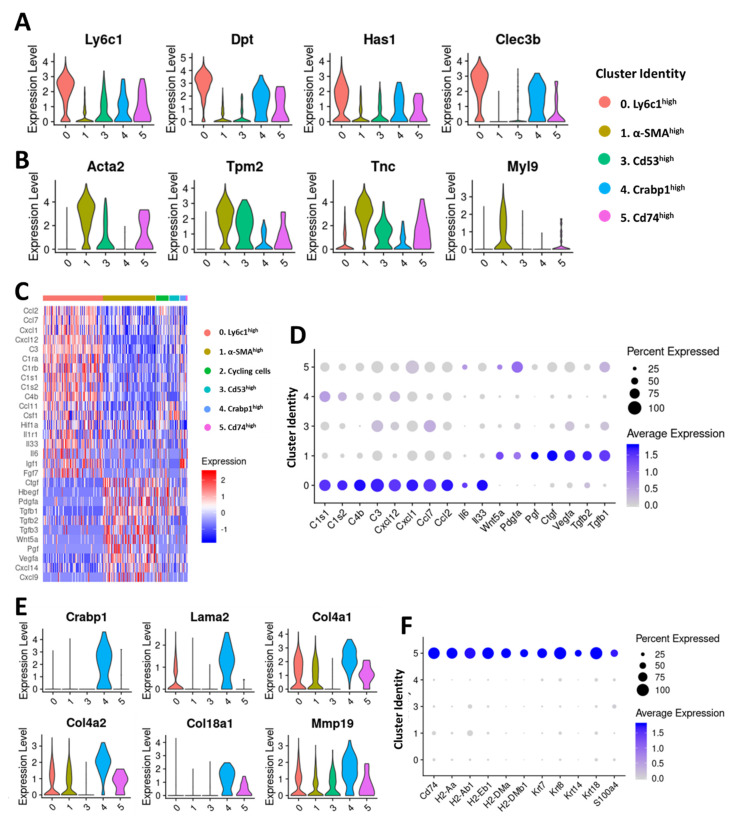
Characterization of CAF clusters in 4T1 mammary tumors. Violin plots showing the expression of Ly6c1^high^ (**A**) and α-SMA^high^ (**B**) cluster markers in all clusters except the dividing/cycling cells. (**C**) Heatmap depicting the expression profiles of growth factors and immune/inflammatory signaling mediators differentially expressed between CAF subpopulations. (**D**) Dot plots showing the expression of selected immune/inflammatory signaling molecules enriched in Ly6c1^high^ CAFs and growth factors enriched in α-SMA^high^ CAFs. Dot size represents the fraction of cells expressing a specific marker in a particular cluster and intensity of color indicates the average expression in that cluster. (**E**) Violin plots showing the expression of selected markers enriched in Crabp1^high^ CAFs. (**F**) Dot plots showing the expression of selected markers of Cd74^high^ CAFs. Dot size represents the fraction of cells expressing a specific marker in a particular cluster and intensity of color indicates the average expression in that cluster.

**Figure 4 cancers-12-01307-f004:**
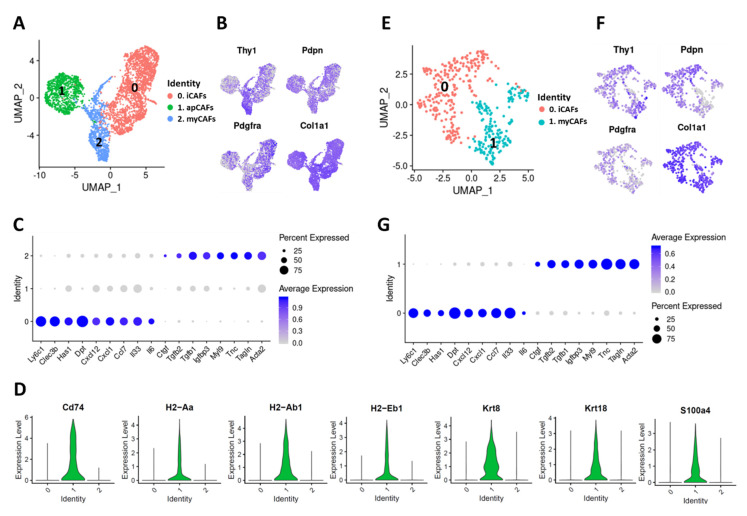
Characterization of CAF subtypes in murine PDAC. (**A**) UMAP plot showing various CAF subtypes identified in PDAC tumors from KPC mice. (**B**) Expression of commonly used CAF markers in KPC-derived PDAC CAFs. (**C**) Dot plot showing the expression of selected markers of Ly6c1^high^ CAFs (iCAFs) and α-SMA^high^ CAFs (myCAFs) in KPC-derived PDAC CAFs. Dot size represents the fraction of cells expressing a specific marker in a particular cluster and intensity of color indicates the average expression in that cluster. (**D**) Violin plots showing the expression of Cd74^high^ CAF (apCAFs) markers in KPC-derived CAFs (cluster identities on X-axis). (**E**) CAF subtypes identified in subcutaneous mT3 tumors. (**F**) Expression of commonly used CAF markers in mT3 tumor-derived CAFs. (**G**) Dot plot showing the expression of selected iCAF and myCAF markers in mT3-derived CAFs.

**Figure 5 cancers-12-01307-f005:**
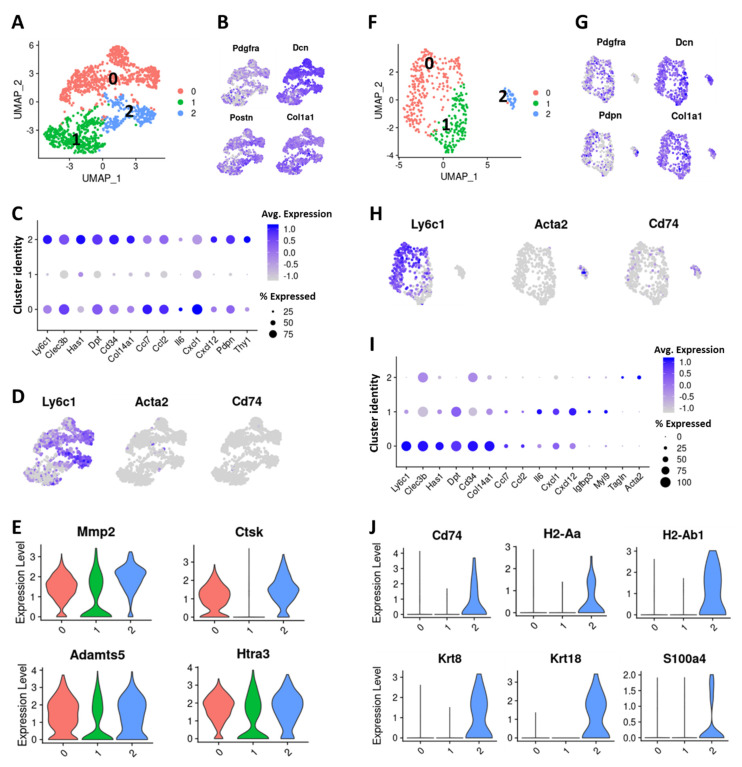
Characterization of normal tissue-resident fibroblasts. (**A**) UMAP plot showing various fibroblast subtypes identified in normal mammary fat pad. (**B**) Expression of commonly used fibroblast/CAF markers in mammary fat pad-derived fibroblast subtypes (**C**) Dot plot showing the expression of selected iCAF markers in mammary fat pad-derived fibroblast subtypes. Dot size represents the fraction of cells expressing a specific marker in a particular cluster and intensity of color indicates the average expression in that cluster. (**D**) Feature plot showing the expression of *Ly6c1*, *Acta2* and *Cd74* in mammary fat pad-derived fibroblasts. (**E**) Violin plot showing the expression of matrix degrading enzymes in mammary fat pad-derived fibroblast subtypes (cluster identity on X-axis). (**F**) UMAP plot showing various fibroblast subtypes identified in normal pancreas. (**G**) Expression of commonly used fibroblast/CAF markers in normal pancreas-derived fibroblasts. (**H**) Feature plot showing the expression of *Ly6c1*, *Acta2* and *Cd74* in pancreas-derived fibroblasts. (**I**) Dot plot showing the expression of selected iCAF markers in pancreas-derived fibroblast subtypes Dot size represents the fraction of cells expressing a specific marker in a particular cluster and intensity of color indicates the average expression in that cluster. (**J**) Violin plots showing the expression of Cd74^high^ CAF (apCAFs) markers in pancreas-derived fibroblasts subtypes.

**Figure 6 cancers-12-01307-f006:**
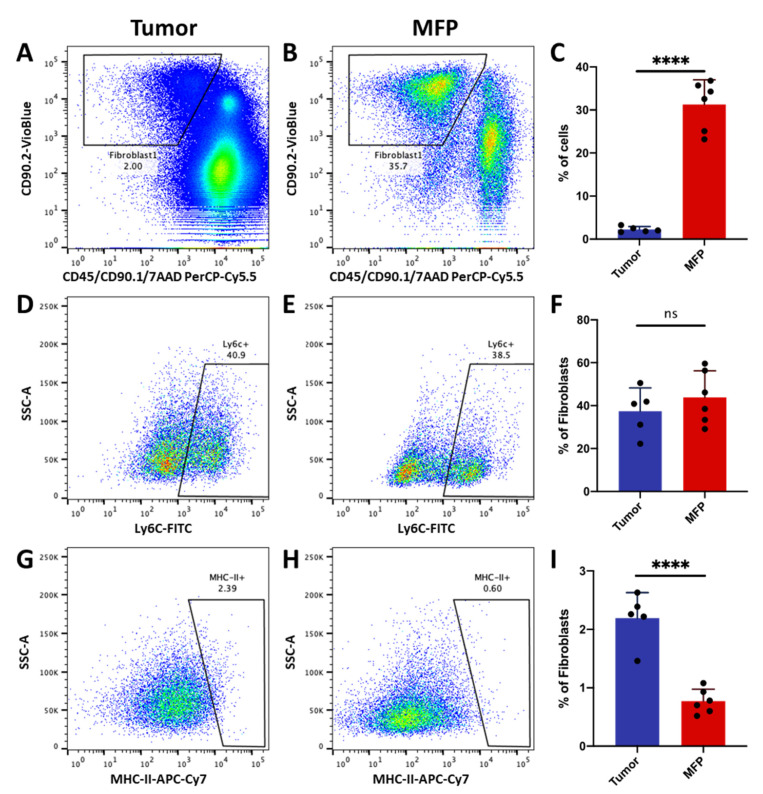
Inflammatory fibroblasts and MHC class II-expressing fibroblasts in mammary tumors and naïve mammary fat pad. Flow cytometry plots identifying viable (7AAD−) CD90.2+/CD90.1−/CD45− fibroblasts derived from (**A**) tumor or (**B**) naïve mammary fat pad (MFP). (**C**) Bar graph of Thy1+ fibroblast abundance in tumor and MFP. Flow cytometry plots identifying Ly6c1^high^ subpopulations from all Thy1^+^ fibroblasts in the tumor (**D**) and (**E**) MFP. (**F**) Bar graph of Ly6c1^high^ subpopulations abundance in the tumor and MFP. Flow cytometry plots identifying MHC class II-expressing subpopulations from all Thy1+ fibroblasts in the tumor (**G**) and (**H**) MFP. (**I**) Bar graph showing the abundance of MHC class II -expressing subpopulation in the tumor and MFP. *n* = 5–6 performed in two independent experiments. Errors bars denote SD. **** *p* < 0.0001, not significant (ns) *p* > 0.005.

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
