# Peer review of "Single-Cell Transcriptomic Analysis of Tumor-Derived Fibroblasts and Normal Tissue-Resident Fibroblasts Reveals Fibroblast Heterogeneity in Breast Cancer"

_cancers, 2020, doi:10.3390/cancers12051307_

Round 1

Reviewer 1 Report

Sebastian et al., indicated three distinct subpopulations including myofibroblastic CAFs, inflammatory CAFs (iCAFs) and antigen-presenting CAFs (apCAFs) existing in both mouse breast and pancreas tumors. Moreover, the inflammatory and antigen-presenting CAF traits were also detected in healthy breast and pancreas tissues, indicating these traits in fibroblasts to be present inherently in benign tissues before being induced in tumors.

This work sounds interesting, novel and timely in this fields. There are only a few minor issues to be revised prior to publication as indicated below.

  1. If definition of myeloid cells is innate immune cells, neutrophils should be included in myeloid cells. However, the authors showed distinct myoid cell populations (1, 2, 3, 5) and neutrophils (8) in Figure 1B. The reviewer did not understand why four distinct myoid cell populations are shown. Should it be merged into one population? The neutrophils should also be included into myoid cell populations.
  2. The authors described the explanation of Figure 7 just after that of Figure 6A-E, by skipping Figure 6F-J in text, causing difficulty for readers to follow. The reviewer suggests that Figure 6F-J should be replaced by Figure 7. In another word, explain iCAF and apCAF traits present in the normal breast tissue in Figure 6 followed by that in normal pancreas tissue in Figure 7 or supplementary Figure. Likewise, the explanation of Figure 3C precedes that of Figure 3B in page 5.  

Author Response

Sebastian et al., indicated three distinct subpopulations including myofibroblastic CAFs, inflammatory CAFs (iCAFs) and antigen-presenting CAFs (apCAFs) existing in both mouse breast and pancreas tumors. Moreover, the inflammatory and antigen-presenting CAF traits were also detected in healthy breast and pancreas tissues, indicating these traits in fibroblasts to be present inherently in benign tissues before being induced in tumors.

This work sounds interesting, novel and timely in this fields. There are only a few minor issues to be revised prior to publication as indicated below.

If definition of myeloid cells is innate immune cells, neutrophils should be included in myeloid cells. However, the authors showed distinct myoid cell populations (1, 2, 3, 5) and neutrophils (8) in Figure 1B. The reviewer did not understand why four distinct myoid cell populations are shown. Should it be merged into one population? The neutrophils should also be included into myoid cell populations.

Each myeloid cluster had a distinct gene expression profile. We have identified these clusters as monocyte-derived cells (expresses monocyte, macrophage and pre-DC markers), macrophages, M2-like macrophages (high expression of M2 markers) and proliferating myeloid cells (high expression of cell cycle genes along with other myeloid markers) (Figure 1B). We have also updated figure 1C with new markers. An in-depth characterization of immune populations is beyond the scope of this manuscript. However, we have added a supplementary table (Table S1) with all the genes enriched in each cluster compared to other clusters for readers who are interested in these immune populations.

The authors described the explanation of Figure 7 just after that of Figure 6A-E, by skipping Figure 6F-J in text, causing difficulty for readers to follow. The reviewer suggests that Figure 6F-J should be replaced by Figure 7. In another word, explain iCAF and apCAF traits present in the normal breast tissue in Figure 6 followed by that in normal pancreas tissue in Figure 7 or supplementary Figure. Likewise, the explanation of Figure 3C precedes that of Figure 3B in page 5.

We have corrected the order in which we reference these figures in the text.

Reviewer 2 Report

Sebastian et al. examined fibroblast heterogeneity in multiple tissues using scRNA-seq. The datasets generated in this study will be of great interest and utility to the research community.  The findings are clearly presented, and the analysis has been carried out according to current technical standards. The main findings from this study largely validate previous studies rather than providing novel insight into fibroblast heterogeneity and in some areas the evidence supporting the authors conclusions could be strengthened. However, this is a highly topical and novel area of research, which requires validation from multiple studies to consolidate our understanding of fibroblast heterogeneity in cancer. The data presented does provide additional tissue/model-specific context to previous studies and is in keeping with Cancers stated aim to avoid unnecessary repetition of experiments. Therefore, I would recommend this paper is accepted for publication, providing that the conclusions are substantiated by further analysis -as detailed below.

The manuscript provides a detailed rationale for analysing fibroblast heterogeneity in breast and pancreatic normal and cancer tissues. The authors use scRNA-seq to examine the phenotype of fibroblasts isolated from orthotopic 4T1 tumours. This consists of two experimental approaches: (1) analysing all 7AAD negative (viable) cells or (2) enriching for fibroblasts through epithelial and immune cell depletion. In the first approach 6420 cells were examined, ~8% of which were identified as CAFs (~500). An under-representation stated to likely be due to the single cell isolation method. Data from this approach is not examined beyond identifying CAFs and adds little information or support to the main conclusions in this paper. The associated dataset will no doubt be useful to other researchers, so this data warrants inclusion. However, the initial two figures are supplementary to the key findings presented and the sections describing these data could be made more concise. The second experimental approach yielded ~1600 fibroblasts, enabling further analysis of subpopulations, which represent the major findings in this study.

The two most prominent clusters/subpopulations identified (iCAFs and myofibroblasts) and CD74+ (antigen presenting CAFs) are consistent with previous findings in PDAC, which is referenced by the authors. This does somewhat compromise the novelty of the data and conclusions presented. Two additional clusters (plus cycling fibroblasts) were identified, which may represent novel fibroblast subpopulations. However, the presented analysis is not sufficient to conclusively determine this. To further support the claim of “we identified six CAF subpopulations” the existence of these subpopulations should be confirmed through further analysis. This could include demonstrating that these clusters are identified using alternative clustering algorithms or varying the number of variable genes used for PCA dimensionality reduction; additionally it should be examined whether these cells are identified in additional datasets (for example the initial analysis of all cells from the 4T1 tumours). Furthermore, it is not clear how many tumours were analysed in each of these datasets and whether these clusters were identified in individual tumours or across multiple tumours. This should be stated in the text and plots overlaying this information on UMAP dimensionality reduction should be added to the figures.

In the remaining sections of the paper the iCAFs/Ly6c1high, myofibroblast/Acta2high and apCAFs/CD74high subpopulations are identified/examined in a combination of novel and publicly available datasets, confirming the presence of iCAF and apCAF (but not myofibroblast) populations in normal tissue, which is further validated by flow cytometry. Although these populations have been described previously the additional tissue/model-specific context that this analysis provides is interesting and potentially valuable to the research community. However, from the analysis performed it is not clear whether the subpopulations identified in these tissues re-capitulate the original phenotypes identified. The authors show that some markers are shared across tissue types/models. For example, identifying Ly6c1 positive populations in normal breast and pancreatic tissues. However, it is not clear whether this constitutes identification of the same phenotype/subpopulation; or merely comparable expression of selected markers. For example, in contrast to the 4T1 tumours, Ly6c1 expression is not limited to a single cluster in Figure 6C and the CD74high cluster is Acta2 positive in normal pancreas (Figure 6H).  To address this the authors could examine whether module scores for the subpopulation markers identified in Figure 3 are uniquely upregulated by clusters in the subsequent analyses; or by aligning the datasets and examining whether these subpopulations cluster together or separately.  

Additional comments:

In Figure S2C the Cd74+ high population is shown to be of fibroblastic origin due to comparable expression of CAF markers. In this figure it would be useful to see the Expression level of these markers in epithelial and immune cells. Furthermore, given that this population is marked by multiple keratins it would also be useful to also show how these markers compare to epithelial cells. Could it be that these cells are a product of (partial-)EMT given the co-expression of keratins and Fsp1?

The sentence starting on Page 8 line 271 should be re-worded. The authors state “Using publicly available scRNA-seq data derived from a genetic mouse model of PDAC, the KPC mouse (Kras+/LSL-G12D ; Trp53+/LSL-R172H; Pdx-Cre) (GSE129455) [15], we identified 3 CAF subtypes (Figure 5A).” This is misleading as these subtypes were identified in the original study by Elyada et al. Amending this sentence to (for example), “we reproduced Elyada et al's findings…” would be more appropriate.

I was unable to access the raw data at the link provided. This data needs to be made available should this manuscript be published.

The markers identified for each subpopulation should also be made available as supplementary tables, preferably in a non-pdf format (e.g. xlsx/csv)

The Flow cytometry axis labels (Figure 7) are quite difficult to read.

Author Response

Reviewer 2

Sebastian et al. examined fibroblast heterogeneity in multiple tissues using scRNA-seq. The datasets generated in this study will be of great interest and utility to the research community.  The findings are clearly presented, and the analysis has been carried out according to current technical standards. The main findings from this study largely validate previous studies rather than providing novel insight into fibroblast heterogeneity and in some areas the evidence supporting the authors conclusions could be strengthened. However, this is a highly topical and novel area of research, which requires validation from multiple studies to consolidate our understanding of fibroblast heterogeneity in cancer. The data presented does provide additional tissue/model-specific context to previous studies and is in keeping with Cancers stated aim to avoid unnecessary repetition of experiments. Therefore, I would recommend this paper is accepted for publication, providing that the conclusions are substantiated by further analysis -as detailed below.

The manuscript provides a detailed rationale for analysing fibroblast heterogeneity in breast and pancreatic normal and cancer tissues. The authors use scRNA-seq to examine the phenotype of fibroblasts isolated from orthotopic 4T1 tumours. This consists of two experimental approaches: (1) analysing all 7AAD negative (viable) cells or (2) enriching for fibroblasts through epithelial and immune cell depletion. In the first approach 6420 cells were examined, ~8% of which were identified as CAFs (~500). An under-representation stated to likely be due to the single cell isolation method. Data from this approach is not examined beyond identifying CAFs and adds little information or support to the main conclusions in this paper. The associated dataset will no doubt be useful to other researchers, so this data warrants inclusion. However, the initial two figures are supplementary to the key findings presented and the sections describing these data could be made more concise. The second experimental approach yielded ~1600 fibroblasts, enabling further analysis of subpopulations, which represent the major findings in this study.

We have merged figure 1 and 2 into one figure to make this section more concise.

The two most prominent clusters/subpopulations identified (iCAFs and myofibroblasts) and CD74+ (antigen presenting CAFs) are consistent with previous findings in PDAC, which is referenced by the authors. This does somewhat compromise the novelty of the data and conclusions presented. Two additional clusters (plus cycling fibroblasts) were identified, which may represent novel fibroblast subpopulations. However, the presented analysis is not sufficient to conclusively determine this. To further support the claim of “we identified six CAF subpopulations” the existence of these subpopulations should be confirmed through further analysis. This could include demonstrating that these clusters are identified using alternative clustering algorithms or varying the number of variable genes used for PCA dimensionality reduction; additionally it should be examined whether these cells are identified in additional datasets (for example the initial analysis of all cells from the 4T1 tumours). Furthermore, it is not clear how many tumours were analysed in each of these datasets and whether these clusters were identified in individual tumours or across multiple tumours. This should be stated in the text and plots overlaying this information on UMAP dimensionality reduction should be added to the figures.

We have repeated the analysis by changing the number of variable genes used for dimensionality reduction and confirmed the presence of six CAF subtypes (Figure S1D-E).

Several studies have characterized myofibroblasts-like CAFs in breast cancer however, ours is the first study characterizing Ly6c1-high inflammatory and Cd74-high antigen presenting fibroblast populations in breast cancer and in normal breast tissue.  In this study, we did not analyze independent biological replicates using scRNAseq however, independent biological replicates (tumors from 5 BALB/C mice and normal mammary fat pads from 6 BALB/C mice mice) were analyzed using flow cytometry to confirm the presence of these populations (figure 6). We have included this information in the ‘methods’.

Another prominent CAF population identified in this study is the Crabp1-high fibroblasts.  Bartoschek et al have previously identified a Crabp1+ fibroblast cluster in MMTV-PyMT mouse model of breast cancer (Bartoschek et al, Nature Communications, 2018). We have re-analyzed this data and have shown that Crabp1-high cluster identified in MMTV-PyMT tumors share markers (Crabp1, Lama, Mmp19, Spon1, Lum etc.) with the Crabp1-high cluster identified in this study (Figure S6). Crabp1 expression was largely restricted to just this cluster in both datasets. These two studies together confirm the existence of Crabp1-high fibroblasts in breast cancer.

We have also identified a small cluster of Cd53-high CAFs 4T1 breast tumors. By aligning this data (4T1-Thy1.1 tumors) to the CAFs identified in the initial analysis (4T1 tumors) we have confirmed the presence of Cd53-high CAFs in both datasets. Although the initial 4T1 scRNA-seq data had only ~500 CAFs, Cd53-high cells from both datasets formed a separate cluster and expressed markers Cd53, Lgals3 and Sdc1 (Figure 6 D-F).

Both Cd53high and Crabp1high CAFs had distinct gene expression profiles. In future studies we will determine whether these CAFs have distinct origin/function or merely represent a transitional state during the differentiation of CAFs.

In the remaining sections of the paper the iCAFs/Ly6c1high, myofibroblast/Acta2high and apCAFs/CD74high subpopulations are identified/examined in a combination of novel and publicly available datasets, confirming the presence of iCAF and apCAF (but not myofibroblast) populations in normal tissue, which is further validated by flow cytometry. Although these populations have been described previously the additional tissue/model-specific context that this analysis provides is interesting and potentially valuable to the research community. However, from the analysis performed it is not clear whether the subpopulations identified in these tissues re-capitulate the original phenotypes identified. The authors show that some markers are shared across tissue types/models. For example, identifying Ly6c1 positive populations in normal breast and pancreatic tissues. However, it is not clear whether this constitutes identification of the same phenotype/subpopulation; or merely comparable expression of selected markers. For example, in contrast to the 4T1 tumours, Ly6c1 expression is not limited to a single cluster in Figure 6C and the CD74high cluster is Acta2 positive in normal pancreas (Figure 6H).  To address this the authors could examine whether module scores for the subpopulation markers identified in Figure 3 are uniquely upregulated by clusters in the subsequent analyses; or by aligning the datasets and examining whether these subpopulations cluster together or separately. 

We have performed a comparative analysis of tumor-derived CAFs and fibroblasts from normal mammary fat pad and found that tumor derived Ly6c1-high cells are highly similar to normal mammary fat pad derived Ly6c1-high cells (cluster 0 in Figure S5). The Ly6c1-high CAFs from tumor aligned well with the Ly6c1-high fibroblasts from normal mammary fat pad and expressed common markers including Ly6c1, Clec3b, Dpt, Ly6a, Htra3, Has1 and Col14a1 (Figure S5A, E). We have also aligned breast- and PDAC-derived CAFs and found that iCAFs, myCAFs and apCAFs in these cell types are highly similar (Figure S4). However, we did observe some tissue/disease state specific differences in gene expression, which was expected.  

Additional comments:

In Figure S2C the Cd74+ high population is shown to be of fibroblastic origin due to comparable expression of CAF markers. In this figure it would be useful to see the Expression level of these markers in epithelial and immune cells. Furthermore, given that this population is marked by multiple keratins it would also be useful to also show how these markers compare to epithelial cells. Could it be that these cells are a product of (partial-)EMT given the co-expression of keratins and Fsp1?

Figure S2D shows that pan-CAF markers such as Pdpn, Pdgfra and Dcn have extremely low expression in immune and epithelial cells.

It is possible that Cd74-high cells are a product of partial EMT. Keratins, claudins and Fsp1 enriched in Cd74+ cells are also highly expressed in epithelial cell (Figure S2D).  A recent manuscript suggests that these Cd74+ cells could have a mesothelial origin (Dominguez et al, Cancer Discovery 2020). In the absence of lineage tracing it will be difficult to conclusively determine the origin of these cells but should be the subject of future studies.

 The sentence starting on Page 8 line 271 should be re-worded. The authors state “Using publicly available scRNA-seq data derived from a genetic mouse model of PDAC, the KPC mouse (Kras+/LSL-G12D ; Trp53+/LSL-R172H; Pdx-Cre) (GSE129455) [15], we identified 3 CAF subtypes (Figure 5A).” This is misleading as these subtypes were identified in the original study by Elyada et al. Amending this sentence to (for example), “we reproduced Elyada et al's findings…” would be more appropriate.

We have made these corrections

I was unable to access the raw data at the link provided. This data needs to be made available should this manuscript be published.

The data has already been uploaded to dryad database and will be made available to the public by the time the manuscript gets published. The uploaded data files are available for download at:

https://datadryad.org/stash/share/01urnKHyOzuvbWbG4pBdpwUker_-jYqbSksrJsE0E3U

The markers identified for each subpopulation should also be made available as supplementary tables, preferably in a non-pdf format (e.g. xlsx/csv)

We have included the markers as supplementary tables (Table S1, Table S2, Table S3)

The Flow cytometry axis labels (Figure 7) are quite difficult to read.

We have increased font size for the axis labels in this figure.

Round 2

Reviewer 2 Report

The authors have done a nice job addressing my previous concerns. My only additional comments regard minor errors found in the supplementary materials.

In Figure S5 the legend is slightly incorrect. It should read:

"C-D) Feature plot showing the expression of iCAF marker Ly6c1 (C) and myCAF marker Acta2 (D) in aligned tumor-derived CAFs and normal mammary fat pad-derived fibroblasts. E) Dot plot showing the expression of selected CAF subtype markers in tumor-derived CAFs and normal mammary fat pad- derived fibroblast clusters."

There is also an unlabelled UMAP plot in the subsequent page of the supplementary figures.

Author Response

Supplementary Figure and figure legend have been corrected